# Evidence of Tri-Exponential Decay for Liver Intravoxel Incoherent Motion MRI: A Review of Published Results and Limitations

**DOI:** 10.3390/diagnostics11020379

**Published:** 2021-02-23

**Authors:** Olivier Chevallier, Yì Xiáng J. Wáng, Kévin Guillen, Julie Pellegrinelli, Jean-Pierre Cercueil, Romaric Loffroy

**Affiliations:** 1Image-Guided Therapy Center, Department of Vascular and Interventional Radiology, François-Mitterrand University Hospital, 14 Rue Paul Gaffarel, BP 77908, 21079 Dijon, France; olivier.chevallier@chu-dijon.fr (O.C.); kguillen@hotmail.fr (K.G.); julie.pellegrinelli@chu-dijon.fr (J.P.); jpcercueil@gmail.com (J.-P.C.); 2Department of Imaging and Interventional Radiology, Faculty of Medicine, The Chinese University of Hong Kong, New Territories, Hong Kong, China; yixiang_wang@cuhk.edu.hk

**Keywords:** diffusion weighted imaging, intravoxel incoherent motion, liver, tri-exponential, b-value

## Abstract

Diffusion weighted imaging (DWI) and intravoxel incoherent motion (IVIM) have been explored to assess liver tumors and diffused liver diseases. IVIM reflects the microscopic translational motions that occur in voxels in magnetic resonance (MR) DWI. In biologic tissues, molecular diffusion of water and microcirculation of blood in the capillary network can be assessed using IVIM DWI. The most commonly applied model to describe the DWI signal is a bi-exponential model, with a slow compartment of diffusion linked to pure molecular diffusion (represented by the coefficient D_slow_), and a fast compartment of diffusion, related to microperfusion (represented by the coefficient D_fast_). However, high variance in D_fast_ estimates has been consistently shown in literature for liver IVIM, restricting its application in clinical practice. This variation could be explained by the presence of another very fast compartment of diffusion in the liver. Therefore, a tri-exponential model would be more suitable to describe the DWI signal. This article reviews the published evidence of the existence of this additional very fast diffusion compartment and discusses the performance and limitations of the tri-exponential model for liver IVIM in current clinical settings.

## 1. Introduction

Diffusion weighted imaging (DWI) sequence has shown high-performance for the detection of malignant liver tumors, particularly liver metastases from solid cancer [1,2,3,4]. There is a growing interest for diffused liver diseases assessment using DWI sequence, particularly for liver fibrosis detection and staging, but also for liver inflammation [5,6,7,8,9,10,11,12]. New treatments might slow or stop the progression of liver fibrosis [13]. For treatment monitoring, a noninvasive method, such as liver DWI, is required. Initially, DWI signal was described as a mono-exponential decay model, owing to a mono-compartmental model of water diffusion [14,15]. In 1986, an additional compartment of diffusion, related to the microcirculation of blood, was described by Le Bihan et al. and the principle of intravoxel incoherent motion (IVIM) was introduced [16,17]. According to IVIM theory, two diffusion compartments can be assessed: a fast compartment of diffusion, related to microperfusion (represented by D_fast_, or D*), and a slow compartment linked to pure molecular diffusion (represented by D_slow_, or D) [17,18]. Therefore, a bi-exponential decay model was used to describe the signal decay. Numerous studies aiming to assess diffused liver diseases using this two-compartment model have been published [7,9,19]. However, high variance in D_fast_ estimates has been consistently shown in literature, restricting its application in clinical practice [9]. This variance could be due to an additional perfusion component [20]. In addition, liver is anatomically and physiologically quite complex, with presence of several vessels types (arteries/arterioles, portal veins/venules), sinusoid capillaries, bile ducts, lymphatic system, and the space of Disse, which is a functionally important intermediate area of exchange between the sinusoids and hepatocytes [21]. Since flowing or moving spins are present in these compartments, which are directly or indirectly connected together, it can be hypothesized that more than two compartments of diffusion exist in the liver and could be analyzed through the recorded DWI signal decay. Recently, evidence of the existence of an additional very fast component of diffusion in the liver has been published [20,22,23,24,25,26]. Therefore, a tri-exponential decay model has been proposed. The aim of this article is to review and discuss the published evidence of a tri-exponential decay behavior of the DWI signal, but also the technical requirements and the limitations of the tri-exponential IVIM model in current clinical settings.

## 2. Classic DWI Mono-Compartmental Model

Molecular diffusion, which is due to microscopic random translational motion of molecules in a fluid, can be studied by a diffusion model [16]. The diffusion coefficient *D* is related to molecular mobility. For pure water, *D* = 2.5 × 10^−3^ mm^2^/s and *D* = 2.3 × 10^−3^ mm^2^/s at 40 and 25 °C, respectively [16,17]. Restricted diffusion phenomenon is observed in fluids when diffusion is restricted to a limited volume, such as intracellular water. A phase shift is produced by the displacement of spins during the time of echo (TE) in the presence of diffusion gradients [16,17]. The loss of phase coherence in the transverse magnetization results in an attenuation of the DWI signal.

Considering mono-compartmental model of diffusion, the DWI signal is regarded as related to the random Brownian motion of water molecule, molecular diffusion. The signal attenuation as a function of b is thus expressed using mono-exponential decay model with the following equation:SI_b_ = SI_0_.exp(−*b*.ADC)(1)
where SI_b_ is the signal intensity at the given *b* value (weighted signal), SI_0_ denotes the signal intensity at b = 0 s/mm^2^ (unweighted signal) which is proportional to e^−TE/T2^, and ADC is the apparent diffusion coefficient.

In the presence of restricted diffusion, ADC is reduced.

## 3. IVIM Theory, Bi-Compartmental Model

Le Bihan et al. demonstrated that the separation of two different anatomical compartments was feasible in the voxel: an intracellular compartment in which molecular diffusion can be assessed, and an extracellular compartment in which the spins’ motions present higher velocities [17].

According to IVIM theory, diffusion is considered bi-compartmental with a fast component of diffusion, related to microcirculation, called pseudo-diffusion, and a slow component of diffusion, linked to pure molecular diffusion. The signal attenuation as a function of b is modeled according to a bi-exponential equation:SI_b_ = SI_0_.[PF.exp(−*b*.D_fast_) + (1 − PF).exp(−*b*.D_slow_)](2)
where PF (also called f) represents the fraction of the pseudo-diffusion compartment, D_fast_ (also called D*) is the pseudo-diffusion coefficient representing the incoherent microcirculation within the voxel (perfusion-related diffusion) and D_slow_ (also called D) is the diffusion coefficient representing the slow (pure) molecular diffusion. In addition, the expression (1-PF) represents the fraction of the molecular diffusion compartment.

D_fast_ values demonstrate high variance between studies, restricting its application in clinical practice. Particularly, D_fast_ is underestimated when few low b-values are included, while it tends to increase when more very low b-values are included [9,27]. D_fast_ value is thus very dependent of the lowest nonzero b-value of the DWI acquisition. Figure 1 shows the dependency of D_fast_ on the low and very low b-values included in the b-value distribution. This suggests that bicompartmental model may not be suitable for the fit of liver DWI.

## 4. Tri-Compartmental Model

The use of a tri-exponential model for IVIM has two goals: first, a better assessment of molecular diffusion by applying a model that is more consistent with the recorded signal decay; second, a separation of two fast diffusion compartments, which are related to extracellular spins’ motions and observed at low and very-low b-values.

### 4.1. Equation

If diffusion is considered as tri-compartmental, the signal attenuation as a function of b is expressed by a tri-exponential equation [22,24]:SI_b_ = SI_0_.[F’_Vfast_.exp(−*b*.D’_Vfast_) + F’_fast_.exp(−*b*.D’_fast_) + F’_slow_.exp(−*b*.D’_slow_)](3)
where D’_fast_ and D’_Vfast_ represent the fast and very fast perfusion-related and pseudo-diffusion coefficients and D’_slow_ represents the molecular diffusion coefficient, thus being similar to D_slow_. F’_slow_, F’_fast_ and F’_Vfast_ are the fractions of each compartment. F’_slow_ is similar to the fraction of the slow diffusion compartment (1—PF), and the combination of F’_fast_ + F’_Vfast_ is similar to PF of the bi-exponential model.

### 4.2. Evidence for Tri-Exponential Decay Model

The presence of multiple perfusion components in the DWI signal has been discussed in several studies [28,29,30]. Recent articles suggest the presence of two perfusion components of diffusion, a very fast compartment of diffusion, represented by the parameters D’_Vfast_ and F’_Vfast_, and a fast compartment of diffusion represented by the parameters D’_fast_ and F’_fast_.

The study of Cercueil et al. aimed to compare mono-, bi- and tri-exponential models in order to determine which of them best fits the IVIM signal of normal liver. In total, 38 and 36 patients for pilot and validation studies were included, respectively [22]. The chosen b-values were 0, 5, 15, 25, 35, 50, 100, 200, 400, 600, 800 s/mm^2^ for the pilot study and 0, 5, 10, 15, 20, 25, 30, 35, 40, 45, 50, 100, 200, 400, 600, 800 s/mm^2^ for the validation study. Since modeling the IVIM signal in a single patient can be challenging, the MR signals of all study participants were averaged together, as it has been previously proposed in order to improve the signal to noise ratio (SNR) [31,32]. Parameters were calculated using a nonlinear least-squares full fitting method. Using the extra sum-of-squares F-test and information criteria to compare the models, Cercueil et al. demonstrated that the tri-exponential model provided a better fit for IVIM signal, with the differences in Akaike information criterion (AIC) and corrected AIC (AICc) scores being strongly in favor of the tri-exponential model.

Using an ingenious adaptive multiexponential IVIM model, Kuai et al. investigated the effect of multiple perfusion components on the measurement of D_fast_ using simulated data [20]. The same adaptive method was used on abdominal DWI acquisitions of 31 volunteers to determine the number of perfusion components in the liver, the spleen, and the kidney [20]. Two perfusion components were found in the liver and the spleen, with larger difference between perfusion components in the liver than in the spleen, whereas only one perfusion component was found in the kidney. However, the kidney presents intermediate IVIM perfusion compared to the liver that presents high IVIM perfusion. The extraction of another perfusion component might be more challenging in the kidney [33]. In addition, the liver and spleen show higher D_fast_ values compared to the kidney [33]. Moreover, regions of interests (ROI) were positioned without discriminating the cortex and the medulla in this study, although these tissues might present different signal decay [23,34]. For the liver and the spleen, the fitting residuals from the multiexponential adaptive IVIM model, with two perfusion components, were significantly smaller than those from the bi-exponential model. Kuai et al. results also implied that the observed high variance of D_fast_ in literature with bi-exponential model could be explained by the presence of multiple perfusion components in the DWI signal [20].

With an extensive DWI protocol including 68 b-values ranging from 0 to 1005 s/mm^2^, a study aimed to assess the number of distinguishable diffusion components in healthy liver, spleen and kidneys [23]. As the T2 relaxation spectra method has been used to assess the multiple components of T2 relaxation in tissues, the computation of apparent diffusion coefficient ‘spectra’ (ADC ‘spectra’) was performed in this study to describe the characteristics and relative contributions of the different diffusion components [35,36]. Eight healthy subjects were included. In addition to the known two components of diffusion proposed by the IVIM model, a third component of diffusion was detected in the liver, in the kidney cortex and medulla, but not in the spleen, where only two diffusion components were observed [23]. Unlike the Kuai et al. study, distinct ROIs were positioned on the kidney cortex and the medulla. This may explain the discrepancy between these two studies concerning the number of detected compartments in the kidney [20,23,34].

Chevallier et al. analyzed the liver DWI signal of 50 scans from 18 volunteers with 16 b-values (0, 3, 10, 25, 30, 40, 45, 50, 80, 200, 300, 400, 500, 600, 700, and 800 s/mm^2^) [24]. Volunteers were scanned twice in a first session, then once in a second session, in order to assess the parameters’ repeatability and reproducibility. For each scan, image series contaminated by evidential physiological motion and other artifacts were manually removed [37]. Data from each scan were then fitted using bi-exponential and tri-exponential decay models, using full fitting and segmented fitting methods. In addition to the IVIM parameters estimation based on individual scans, the measured signals at each b-value from the 50 scans were additionally averaged together; the averaged signals (total-averaged) were then fitted with the four approaches as they were from a single scan [22,32]. Adjusted R^2^, extra-sum-of-squares F-test and AICc were used for model and fitting method comparison. Overall, for the individual scans, the tri-exponential model was favored over the bi-exponential model. Among the 50 individual scans, tri-exponential model was favored for 44 to 50 scans with F-test and for 42 to 49 scans with AICc (depending on the fitting methods). The same finding was observed with the total-averaged fits, with AICc analysis strongly suggesting that bi-exponential model was unlikely to be correct compared to tri-exponential model. For the total-averaged data fits, according to the evidence ratio, tri-exponential model with full fitting method was 1.60 × 10^11^ times more likely to be correct than bi-exponential model with full fitting method. Figure 2 shows the fitting curves obtained with the four methods and total-averaged data. In addition, graphical analysis of the residuals of the fits versus each b-value demonstrated that the tri-exponential model presented smaller and more randomly distributed residuals compared with the bi-exponential model (Figure 3). With the bi-exponential model, the residuals were not randomly scattered showing systematic patterns, therefore suggesting autocorrelation in the residuals and that the bi-exponential model may be unsuitable [38].

In a study aiming at investigating the dependency of IVIM parameters on the used B_0_ field strength, the tri-exponential model was explored [25]. With 20 healthy volunteers, the DWI signal of the liver was fitted with bi-exponential and tri-exponential models. Numerous very low b-values images were acquired (14 b-values < 5 s/mm^2^). AICc were calculated for model comparison and the probability that the tri-exponential was more appropriate was larger than 99.999% for all volunteers [25].

Another study was conducted in order to find optimized b-values that reduce the fit uncertainty in all tri-exponential parameters [26]. With simulated data, AIC comparison favored the bi-exponential model if the number of b-values was small, whereas it favored the tri-exponential model if more b-values were added to the b-values distribution. In addition, when the SNR increased, less b-values were required to favor the tri-exponential model. AIC comparison also indicated that the tri-exponential model was better suited to describe the data than the bi-exponential model for in vivo measurement with three volunteers.

Characteristics of the studies comparing tri-exponential and bi-exponential models for liver IVIM are shown in Table 1.

As mentioned previously, other abdominal organs might also demonstrate a tri-exponential decay behavior of the DWI signal [20,23,34]. In a recent study, authors tested the incorporation of a third component of diffusion in the IVIM model for kidney tissue assessment, since there is both perfusion of the blood vessels and flow of pre-urine through the tubuli and collecting ducts in the kidney [39]. With a DWI protocol including 16 b-values (0, 2, 4, 8, 12, 18, 24, 32, 40, 50, 75, 110, 200, 300, 450, 600 s/mm^2^) and a group of eight healthy volunteers that were scanned at baseline and during three increasing doses of continuous intravenous Angiotensin II infusion, this study demonstrated that tri-exponential analysis was able to detect changes in renal perfusion during pharmacologically induced renal perfusion modulation. Parameter F’_fast_ correlated with the glomerular filtration rate, while there was an inverse correlation between the parameter F’_Vfast_ and the renal plasma flow [39]. Note that both coefficients D’_fast_ and D’_Vfast_ were fixed in order to improve the robustness of the model. Therefore, tri-exponential model for IVIM might also be preferable for other abdominal organs’ assessments. 

### 4.3. Tri-Exponential Model Fitting Methods.

As for bi-exponential fitting, full fitting [22,24] and segmented fitting [24,25,26] methods have been used for the tri-exponential model. The full fitting method was performed using nonlinear least squares regression model-based algorithm [22,24]. By definition, the sum of the fraction of each compartment is equal to 1 (F’_slow_ + F’_fast_ + F’_Vfast_ = 1). Therefore, tri-exponential function can be reduced to five rather than six parameters to solve [24,25,26]. For instance, the parameter F’_Vfast_ can be removed from the equation, leading to:SI_b_ = SI_0_.[(1 − F’_fast_ − F’_slow_).exp(−*b*.D’_Vfast_) + F’_fast_.exp(−*b*.D’_fast_) + F’_slow_.exp(−*b*.D’_slow_)](4)

Segmented fitting method consists of a two-step approach. It is commonly accepted that only the slow diffusion compartment contributes significantly to the DWI signal for b-values higher than 100–200 s/mm^2^ [9,24,25,26,37,40,41,42,43]. Therefore, the estimation of D’_slow_ can first be obtained by a linear fit using only b-values above a certain b-value threshold (generally, 100–200 s/mm^2^). Second, the obtained D’_slow_ can be substituted into Equation 4 and a nonlinear regression using all b-values allows the estimate of the other four parameters [25,26]. Moreover, after the first step of segmented fitting method, F’_slow_ can be estimated as by extrapolation of the curve at b = 0 s/mm^2^, leading to calculate F’_slow_ as SI_int_/SI_0_ where SI_int_ is the b = 0 s/mm^2^ intercept of the fit [37,41,43]. Therefore, both parameters D’_slow_ and F’_slow_ can be substituted into Equation 4, reducing the number of unknown parameters to three rather than four for the nonlinear fit [24].

In the Chevallier et al. study, while full fitting method was slightly favored versus segmented fitting method in terms of fit performance, the differences remained negligible [24]. However, scan–rescan repeatability and reproducibility of parameters were better with the segmented fitting method than with the full fitting method, apart from D’_Vfast_ for which the results were similar [24]. In addition, for both fitting methods, starting points and boundary constraints can be introduced to improve the fit [24,25,26,37,40].

The calculation of the parameters in an adaptive manner has also been explored using an adaptive multiexponential IVIM model [20]. New perfusion components are added gradually to the equation, depending on the signal attenuation ratio at b-values inferior to b-values related to the last diffusion compartment that was detected. This method permits to assess the number of perfusion components and the calculation of the parameters in a progressive manner.

A quite different method has been proposed, that consists of fitting the DWI signal in a non-negative least-squares manner to a distribution of decaying exponential function using minimum-amplitude energy regularization [23]. From the resulting ADC spectra, the diffusion coefficients can be extracted. Based on our understanding, the main goal of this method is to identify the number of diffusion components in abdominal organs.

All these studies used a ROI-based analysis method, which offers better estimation than pixel-fitting method when the SNR is low [44,45].

Other methods that introduced a spatially constrained incoherent motion model of the DWI signal decay and an efficient iterative “fusion bootstrap moves” solver have been proposed and might improve the accuracy and robustness of IVIM parameters estimates [46,47]. Bayesian fitting method has been tested for the bi-exponential model and may lead to a more robust estimation [48,49,50]. A novel segmented Bayesian method was also proposed and explored on simulated images and real data of patients with head-and-neck and rectal cancers [51]. This last method was also studied in combination with spatial regularization through a conditional autoregressive prior specification and has shown promising results [51]. Recently, neural-network-based fit approaches have been tested for bi-exponential IVIM with interesting results [52,53]. However, to our knowledge, none of these algorithms have been tested for the tri-exponential model. Regarding the positive results of Bayesian and artificial neural-network based methods for bi-exponential IVIM model, it is very likely that they would improve the accuracy and robustness of tri-exponential parameters.

Based on the assumption that the very fast compartment of diffusion is related to the presence of large blood vessels in the ROI that are not visually identified, other authors considered the very fast-decaying component in the DWI signal as a blood-flow rather than a tissue perfusion [54]. They proposed a new algorithm that considers spatial constraints on a tri-exponential IVIM model dealing simultaneously with the all voxels in the ROI. This method allows an automatic detection of this very fast component of diffusion and provides both the spatial distributions of all the IVIM parameters over the defined ROI and the parameters estimates [54,55].

### 4.4. Tri-Exponential Parameters Values in Healthy Liver

Published values of tri-exponential parameters are shown in Table 2. Main acquisition parameters for DWI sequence by studies are shown in Table 3. Overall, four studies demonstrate similar diffusion coefficients values, with D’_Vfast_ ranging from 380 to 500 × 10^−3^ mm^2^/s, D’_fast_ ranging from 16 to 26.5 × 10^−3^ mm^2^/s, D’_slow_ ranging from 0.98 to 1.35 × 10^−3^ mm^2^/s [20,22,24,26]. D’_slow_ is consistent with literature reports and bi-exponential model [9]. In a previous study where the lowest b-value was b = 10 s/mm^2^, thus reducing the tri-compartmental model to a bi-compartmental model, the D_fast_ value was 12.34 × 10^−3^ mm^2^/s [12]. Although it is slightly lower than the D’_fast_ values previously reported, it remains in the same order of magnitude, suggesting the potential accuracy of those previous D’_fast_ values.

Lower D’_Vfast_ and larger D’_fast_ were found in one study [23]. This can be explained by the lake of very low b-value inferior to b = 15 s/mm^2^. The contribution to the DWI signal of the very fast compartment of diffusion is indeed very low [20,22,24]. It was estimated at 11.7% and 13.5% at b = 0 s/mm^2^, before dropping dramatically to 3.4% at b = 3 s/mm^2^ or 2.09% at b = 5 s/mm^2^, to finally become negligible at b = 10 s/mm^2^ with a contribution to signal of 0.15% or 0.29% [22,24]. This highlights the importance of an adequate sampling of the DWI signal decay for very low b-values, ranging from b = 0–10 s/mm^2^. Indeed, most of the articles that explored the tri-exponential model presented subsampling of the very low b-values, with only 0–2 b-values between b = 0 and b = 15 s/mm^2^ (Table 3) [20,22,23,24]. On the contrary, with 15 b-values in the range b = 0–15 s/mm^2^, several very low b-values between b = 0 and b = 1 s/mm^2^ and between b = 1 and b = 5 s/mm^2^, one study found much larger D’_Vfast_ and D’_fast_, i.e., 2453 and 81.3 mm^2^/s, respectively (Table 2 and Table 3) [25]. The authors suggested those larger values could be explained by the smaller minimal b-values used for the acquisitions. However, another recent study aiming to find an optimized b-value distribution for reproducible tri-exponential IVIM in the liver, also included several very low b-values in an optimized b-values data set [26]. Despite this b-values distribution, lower D’_Vfast_ and D’_fast_, i.e., 500 and 16 mm^2^/s, were found in three volunteers. These last coefficients are much closer to those published in previous studies, particularly D’_fast_ [20,22,24]. The parameter variations between these two studies could be explained by the different slice orientation that has been used for acquisition, sagittal in the first [25], and axial in the second [26]. The presence of different inflow effects secondary to the use of respiratory triggering in one study compared to free breathing in the other could be another explanation. Inclusion of larger vessels within the ROI might have been produced by free breathing.

In the Chevallier et al. study, parameters were calculated individually for the 50 DWI scans of 18 volunteers [24]. In addition to the parameters’ estimation based on the individual scans, the measured signals at each b-value from the 50 scans were also additionally averaged together in order to increase the SNR [22,24,31,32]. The averaged signals (total-averaged) were then fitted. The mean D’_fast_ value obtained from the individual scans was similar to D’_fast_ value calculated from the fit of the total-averaged 50 scans signals. On the contrary, the mean D’_Vfast_ value estimated from the individual scans was much larger than D’_Vfast_ calculated from the total-averaged data. This underlines the importance of an adequate SNR at very low b-values. In addition, very low b-values range was subsampled in this study (Table 3).

F’_slow_ value appears quite similar between all studies, ranging from 68.7% to 76.1%, thus the fraction of the perfusional components (F’_Vfast_ + F’_fast_) is also quite similar. Since F’_Vfast_ + F’_fast_ is similar to PF and F’_slow_ = 1 − (F’_Vfast_ + F’_fast_), the difference in F’_slow_ between studies can be partially explained by the different TE that were used (Table 2 and Table 3). Overall, both perfusion fractions, F’_Vfast_ and F’_fast_ are very close to each other.

### 4.5. b-Value Distribution for Tri-Exponential Model

As it has been discussed previously for the IVIM bi-exponential model, the choice of an optimal b-values distribution is crucial for accurate parameters estimation and to reduce the fit uncertainty [9,33]. An adequate sampling of b-value related to each diffusion compartment is essential. For b-values ranges 0–10 or 0–15 s/mm^2^, the three compartments contribute to the DWI signal [20,22,24]. For 10–15 ≤ b ≤ 100–200 s/mm^2^, the contribution to signal of the very fast diffusion compartment becomes negligible, while the fast and the slow diffusion compartments contribute to the DWI signal. For b ≥ 100–200 s/mm^2^, only the slow diffusion compartment contributes to the signal. Therefore, for a b-values set, we can hypothesize that the number of b-values b ≤ 10–15 s/mm^2^ should be the largest, followed by the number of b-values 10–15 s/mm^2^ ≤ b ≤ 100–200 s/mm^2^, with the number of b-values b ≥ 100–200 s/mm^2^ being the smallest. An optimized b-values set, including numerous very low b-values, has been proposed and the acquisition of a minimum of 16 b-values is recommended [26]. In addition, regarding the very low contribution to the signal of the very fast compartment, and to a lower extent the contribution of the fast compartment, increasing the number of excitations for very low b-values seems necessary.

### 4.6. Effects of the B_0_ Field Strength and the Time of Echo on Tri-Exponential IVIM Parameters

Effects of the B_0_ field strength on IVIM parameters has been previously reviewed [9]. The parameters D_slow_ and PF show, respectively, a decrease and an increase with increasing B_0_ strength [9]. Dependency of the tri-exponential parameters on the magnetic field B_0_ has been recently studied [25]. With 20 volunteers scanned at 1.5T and 3T with 24 b-values, the median of D’_Vfast_ and D’_fast_ decreased slightly with increased field strength. In contrast, a significant dependency of D_fast_ on B_0_ was found. Nevertheless, dependence of D’_Vfast_ and D’_fast_ on B_0_ could have not been detected because of the high fit uncertainty. In addition, no significant dependency was found on B_0_ for the fraction parameters, F’_Vfast_, F’_fast_ and F’_slow_. Consistent with literature, D’_slow_ showed decrease with increasing field strength [9].

Blood and liver tissue present distinct T2 values. The T2 of liver tissue is significantly shorter than the T2 of blood. It has been shown that there is a dependency of PF on the TE, with significant increase of PF when TE is prolonged [32]. Therefore, an increase of F’_Vfast_ and/or F’_fast_ and a decrease of F’_slow_ are expected with increasing TE. This is the trend that we can observe in Table 2 and Table 3. A short TE value of 50–70 ms is generally used in IVIM studies. The use of shorter or longer TE could thus produce discrepancy between studies.

### 4.7. Origins of the Fast and Very Fast Compartments of Diffusion in Liver

The origins of the two perfusion components of diffusion remain not fully understood. The hepatic perfusion is indeed complex. The liver is irrigated by a unique dual-blood supply with 75–80% of hepatic flow coming from portal vein and 20–25% from hepatic artery [56,57]. It contains several vessel types, such as arteries/arterioles, portal veins/venules, hepatic veins/venules, and sinusoid capillaries. In addition, the sinusoid capillaries present zonal gradients of blood flow velocity [58]. Various regimes of blood flow can also be present in vessels [30,59]. An admixture of various perfusion components in liver parenchyma has already been suggested and referred to as the multiple-perfusion-components theory [29]. Regarding the coefficient values of very fast and fast compartments, it has been hypothesized that the fast compartment reflects microperfusion effects, whereas the very rapid flowing spins of the very fast compartment are probably located in larger vessels (arterioles and/or portal venules and/or hepatic venules) [22]. Regarding the contributions to the signal of the diffusion compartments and the diffusion coefficients values that Wurnig et al. found in their study, the authors suggested that the fast and very fast diffusion compartments might be interpreted as portal and arterial blood pools [23]. However, the ratio F’_Vfast_/F’_fast_ has been compared to the estimated relative signal contribution of arterial and venous blood (S_A/V_) based on the T2 decay [25]. As F’_Vfast_/F’_fast_ did not show the field dependence of S_A/V_, it seems unlikely that F’_Vfast_ and F’_fast_ represent arterial perfusion and portal venous blood compartments [25]. The authors suggested that the tri-exponential “model” should be rather regarded as a tri-exponential “representation” and that the tri-exponential behavior originates from a distribution of flow velocities due to the presence of different compartments and different vessel sizes [25,60].

Since the slow compartment of diffusion is related to intracellular molecular diffusion [16], faster components of diffusion are related to extracellular moving spin. The extracellular space includes vascular and interstitial compartments. Moreover, spins’ velocities between the three observed diffusion compartments present a magnitude ratio of about 10 to 30, and even higher regarding D’_Vfast_ values in some studies. This large difference in velocities suggests that three very different anatomical compartments contribute to the DWI signal. In addition, the liver presents an important interstitial compartment, called the space of Disse, in which moving spins are likely to contribute to the DWI signal. Therefore, we can hypothesize that the very fast component of diffusion is associated with microcirculation, whereas the fast component of diffusion is linked to moving spins within the interstitial compartment. Further studies are required to determine the origins of the perfusion components of diffusion.

### 4.8. Limitations of the Tri-Exponential Model

The tri-exponential model demonstrates high fit uncertainty [24,25]. The low SNR of DWI imaging and the low contribution to signal of the fastest compartments of diffusion can partially explain this instability. With an artificial increase of the SNR by averaging the data of several subjects, the fit uncertainty can be reduced with lower SD for parameters [22]. However, these SDs remain high for the two fastest diffusion compartments [22]. Perfusion related parameters, D’_Vfast_, D’_fast_, F’_Vfast_, F’_fast_ demonstrate even worse stability at individual level [24]. It has been reported that an increased number of b-values results in a more precise estimation of the parameters [26,43,61]. As discussed previously, it is also crucial to include enough very low b-values in the b-values distribution. However, extensive b-values images require long acquisition time that might not be applicable for daily clinical applications. In addition, some MRI platforms present fixed b-values or increments, thus not allowing the selection of an optimal b-value distribution for tri-exponential IVIM. Moreover, the generation of parametric IVIM maps, which might be preferred for clinical application, would even be more challenging than the ROI analysis method that has been performed.

The assessment of the very fast compartment of diffusion is challenging. Wang et al. investigated a combined use of bi-exponential IVIM parameters for liver fibrosis evaluation [12]. Interestingly, only b-values ≥ 10 s/mm^2^ images have been acquired. As the contribution to signal of the very fast compartment of diffusion becomes negligible at b = 10 s/mm^2^, the acquisition of only b-values ≥ 10 s/mm^2^ reduces the tri-compartmental model to a bi-compartmental model. Using a three-dimensional tool that included all three bi-exponential IVIM parameters, results were particularly compelling to differentiate patients without liver fibrosis and patients with F1 or F2 fibrosis (AUC, respectively, 0.986 and 1) in cases of viral hepatitis. Therefore, the assessment of the very fast compartment of diffusion may not be essential for liver disease evaluation [12]. Moreover, using the bi-exponential model for b-values ≥ 10 s/mm^2^ might reduce the fit uncertainty. Indeed, less parameters require to be calculated and the signal contamination of the very fast compartment is negligible for b ≥ 10 s/mm^2^. Other authors proposed the removal of voxels within the ROI presenting tri-exponential decay [62]. To this end, these voxels were identified on an ADC image created from b = 0 and b = 10 s/mm^2^. Similarly, Gambarota et al. proposed to first detect the compartment number by using the non-negative least squares method, and then to process the fit without the b = 0 data point in pixels presenting a tri-exponential decay [63]. However, no b-values between b = 0 and b = 10 s/mm^2^ were included in these last two studies. If several very low b-values ≤ 10 s/mm^2^ images are acquired, we believe that the majority of voxels will present a tri-exponential decay (Figure 4) [64].

If the bi-exponential model is processed with only nonzero b-values, b = 0 and very low b-values images can still be useful. Recently, the concept of diffusion-derived vessel density (DDVD) parameter has been proposed to address the initial fast signal decay [65,66,67]. This concept proposed that the relationship between liver DWI signal and b-value can be separated into two parts: part-1 is the signal difference between the b = 0 s/mm^2^ image and the first very low b-value image (usually b = 1 or 2 s/mm^2^ image); the rest is part-2 and fitted with a bi-exponential decay model. On DWI images, blood vessels show a high signal on the unweighted DWI images and low signal with the application of diffusion gradient, even at very low b-value (e.g., 1–15 s/mm^2^). Therefore, the difference between these unweighted and weighted images would reflect the extent of tissue vessel density (referred to as DDVD) [65,66,67]. DDVD analysis could represent a potential biomarker for detecting liver fibrosis. Moreover, the combination of DDVD and bi-exponential IVIM parameters has been shown to improve the separation of fibrotic and nonfibrotic livers [65].

More than two perfusion components of diffusion may also exist and would require another model [20,29,30]. However, it is unlikely that other fast diffusion compartments may be assessed with the present hardware. Moreover, an additional slow component of diffusion has been explored but requires higher b-values to be assessed [61,68]. The addition of another diffusion compartment to the proposed tri-exponential model would make the equation more complex and decrease the fit certainty.

### 4.9. Clinical Applications

To our knowledge, this tri-exponential model with two fast components of diffusion has not been tested for the assessment of diffuse liver disease or liver lesion. The adequacy of the tri-exponential model in diseased settings requires further investigation.

However, the tri-exponential model has been tested for the assessment of renal masses [69]. In this study, parameters obtained from diffusion tensor imaging parameters, IVM bi-exponential model (D_slow_, PF) and tri-exponential model (D’_slow_, F’_slow_, F’_fast_, F’_Vfast_) were compared for the characterization of renal lesions in 16 patients. Histological examination after surgery allowed the determination of tumor type and specificities. Parameters values were found to be representative of some histological features. Both parameters PF and F’_Vfast_ seemed to correlate to vascularization. However, the tri-exponential model provided additional information over the bi-exponential model, since only the tri-exponential parameters F’_fast_ and F’_Vfast_ demonstrated significant differences between different tissues whereas the bi-exponential parameter PF did not [69]. These results highlight the potential improvement in tumor assessment that is provided by the tri-exponential model.

## 5. Conclusions

This review demonstrates that there is concrete evidence of the presence of another fast component of diffusion in the healthy liver. The tri-exponential decay model better fits the recorded DWI signal than the classic bi-exponential IVIM model. However, tri-exponential model application remains challenging. In typical clinical settings, the parameters related to the very fast compartment and fast compartment of diffusion show apparent instability with high fit uncertainty. An extensive DWI-imaging protocol including a high number of very low b-values is required. Technical improvements and more robust fitting methods might allow a more accurate estimate of the tri-exponential IVIM parameters. Further studies are required to explore the performance of the tri-exponential IVIM model in the assessment of liver diseases. New regression models, which allow a better assessment of the microcirculation in the liver, are desired.

## Figures and Tables

**Figure 1 diagnostics-11-00379-f001:**
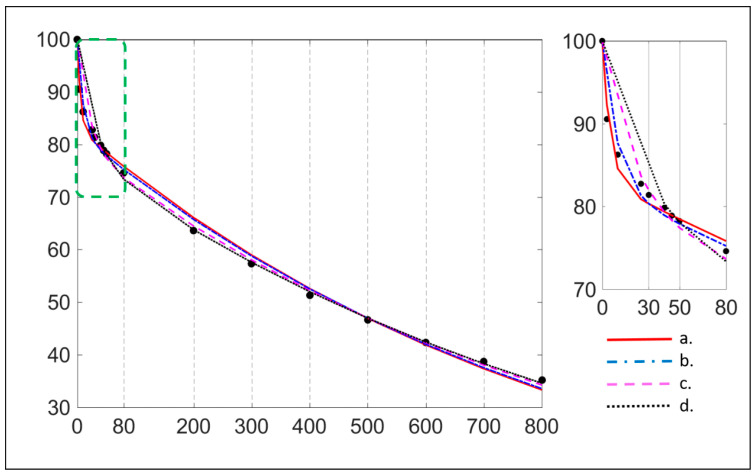
Comparison diffusion weight imaging (DWI) signal fits with the bi-exponential model (full fitting method) using 16 b-values (0, 3, 10, 25, 30, 40, 45, 50, 80, 200, 300, 400, 500, 600, 700, 800 s/mm^2^) (a) and after the successive removal of very low and low b-values (b, c, d). After the successive removal of very low and low b-values, the initial slope of the fitted-curve becomes less and less steep; the D_fast_ value therefore decreases. Notice that the D_slow_ value also decreases, while the PF value increases. (b), b = 3 s/mm^2^ removed; (c), b = 3, 10 s/mm^2^ removed; (d), b = 3, 10, 25, 30 s/mm^2^ removed; D_fast_ = 193.6, 103.6, 48.1, and 35.2 × 10^−3^ mm^2^/s, D_slow_ = 1.14, 1.12, 1.06, and 1.02 × 10^−3^ mm^2^/s, and PF = 16.9%, 17.7%, 20.3%, and 21.8%, for (a); (b); (c), and (d), respectively.

**Figure 2 diagnostics-11-00379-f002:**
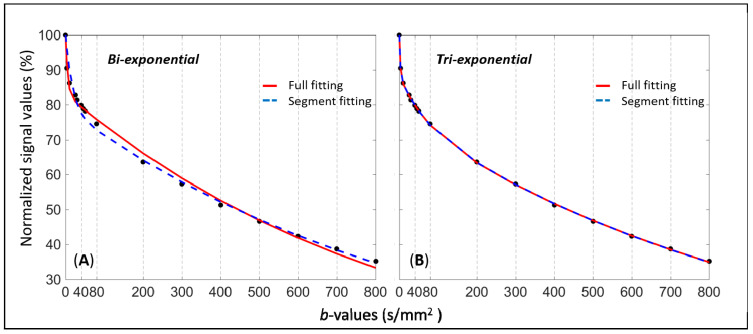
A comparison of signal fitted curves using bi-/tri-exponential models with full or segmented fitting (b-value threshold = 200 s/mm^2^). (**A**) For the bi-exponential model, both fittings do not fit well the initial part of the diffusion signal decay. (**B**) For the tri-exponential model, both fittings show a good fit of diffusion signal decay, with the two fitted curves almost indistinguishable.

**Figure 3 diagnostics-11-00379-f003:**
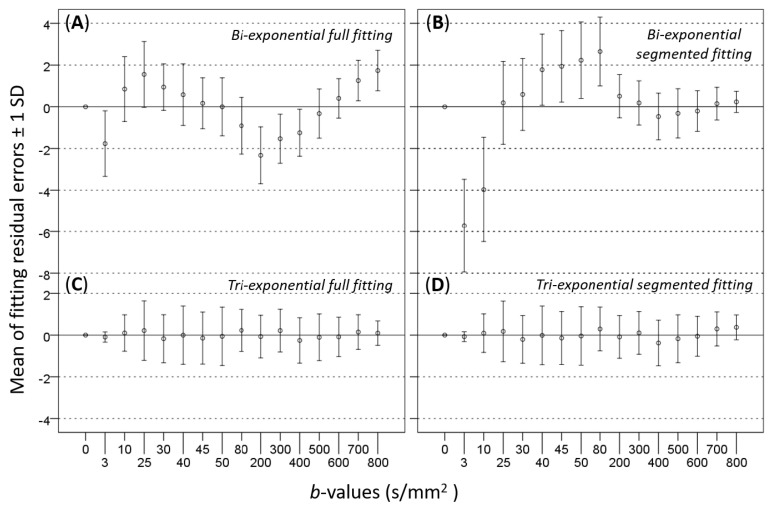
Residuals distribution from the fits of 50 individuals DWI scans using bi-/tri-exponential models with full or segmented fitting (b-value threshold = 200 s/mm^2^). Error bars: mean ± SD (in %, *n* = 50 scans). Bi-exponential model (**A**,**B**) shows strong errors in predicted signal. The residuals in (**A**,**B**) are not randomly scattered showing systematic patterns, therefore suggesting autocorrelation in the residuals. Tri-exponential models (**C**,**D**) show smaller and more random distribution of the residuals.

**Figure 4 diagnostics-11-00379-f004:**
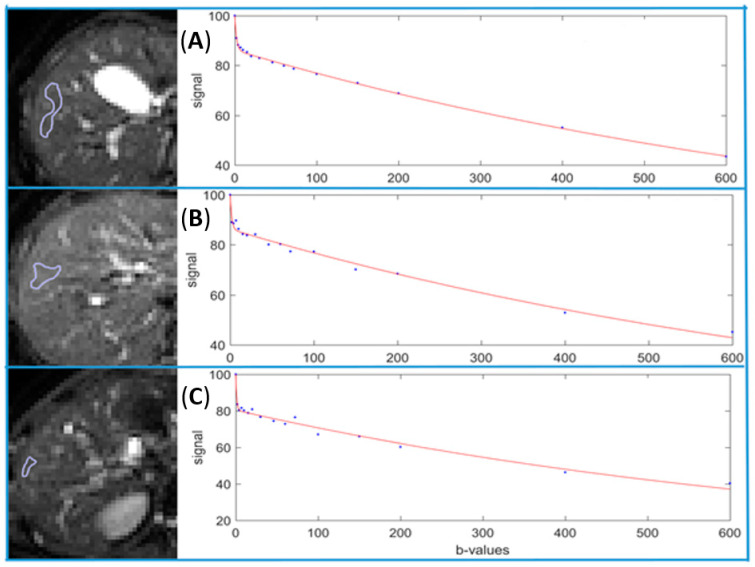
Bi-exponential full fitting curves of three portions of liver parenchyma from three healthy livers. On b = 0 images, small ROIs are drawn on three different location on the liver parenchyma excluding bright pixels which would contain a ‘visible’ vessel (**A**–**C**). The b-value distribution is 0, 2, 4, 7, 10, 15, 20, 30, 46, 60, 72, 100, 150, 200, 400, 600 s/mm^2^, and fitting starts from b = 0 image. Note although the ROIs do not contain a visible vessel, a steep drop of signal from b = 0 to b = 2 s/mm^2^ can still be seen for (**A**–**C**), this would be caused by subpixel microvessels which show high signal on b = 0 image while low signal on b = 2 image.

**Table 1 diagnostics-11-00379-t001:** Characteristics of the studies comparing tri-exponential and bi-exponential models for liver IVIM.

Authors, Year	No. of Subjects	No. of Scans	Fitting Comparison Methods
Cercueil et al., 2015 [22]	38 ^#^	38 ^#^	Extra sum-of-squares F-test and AIC, AICc
	36 ^##^	36 ^##^	
Kuai et al., 2017 [20]	31	31	Fitting residuals data
Chevallier et al., 2019 * [24]	18	50	Adjusted-R^2^ Extra sum-of-squares F-test AIC, AICc Graphical analysis of the residuals
Riexinger et al., 2019 ** [25]	20	40	AICc
Riexinger et al., 2021 [26]	3	3	AIC

No, number; AIC, Akaike information criterion; AICc, corrected Akaike information criterion. * volunteers had their liver scanned twice in the same session and then once in another session. Four scans were excluded due to bad quality of the images. ** abdominal data of volunteers were acquired in two consecutive measurements at 1.5T and 3T. ^#^ pilot study; ^##^ validation study.

**Table 2 diagnostics-11-00379-t002:** Reported values of tri-exponential parameters in healthy liver by studies.

Authors, Year	D’_Vfast_	D’_fast_	D’_slow_	F’_Vfast_	F’_fast_	F’_slow_
Cercueil et al., 2015 ^a^ [22]	391.96	19.54	1.23	17.1	17.6	65.3
Cercueil et al., 2015 ^b^ [22]	404.00	26.50	1.35	13.5	13.7	72.7
Kuai et al., 2017 [20]	386.25	19.32	1.21	17	17	66
Wurnig et al., 2018 [23]	270	43.8	1.26	13.4	7.8	73.8
Chevallier et al., 2019 ^c^ [24]	448.8	15.4	0.98	11.7	11.7	76.6
Chevallier et al., 2019 ^d^ [24]	1911.2	16.1	0.98	11.7	11.9	76.4
Riexinger et al., 2019 ^e^ [25]	2453	81.3	1.22	15.2	16.1	68.7 *
Riexinger et al., 2019 ^f^ [25]	2333	65.9	1.00	15.9	17.4	69.7 *
Riexinger et al., 2021 ^g^ [26]	500	16	1.1	10.8	13.1	76.1 *

^a^ pilot study; ^b^ validation study; ^c^ results from averaged data from 50 scans with segmented fitting method; ^d^ mean parameter values of the 50 scans calculated one by one with segmented fitting method; ^e^ results at 1.5T; ^f^ results at 3T; ^g^ in vivo measurement.* F’_slow_ calculated from reported F’_Vfast_ and F’_fast_ as F’_Vfast_ + F’_fast_ + F’_slow_ = 100%; D’_Vfast_, D’_fast_ and D’_slow_ in 10^−3^ mm^2^/s; F’_Vfast_, F’_fast_ and F’_slow_ in %.

**Table 3 diagnostics-11-00379-t003:** Main acquisition parameters for DWI sequence by studies.

Authors, Year	B_0_	TE	Breathing Management	No. of b-Values	b-Values ≤ 15	Highest b-Value
Cercueil et al., 2015 ^a^ [22]	3T	68 ms	NET	11	0, 5, 15	800
Cercueil et al., 2015 ^b^ [22]	3T	67 ms	NET	16	0, 5, 10, 15	800
Kuai et al., 2017 [20]	3T	68 ms	NET	11	0, 5, 15	800
Wurnig et al., 2018 [23]	3T	57 ms	FB	68	0, 15	1005
Chevallier et al., 2019 [24]	3T	55 ms	RT*	16	0, 3, 10	800
Riexinger et al., 2019 [25]	1.5T	100 ms	FB	24	0.2, 0.4, 0.7, 0.8, 1.1, 1.7, 3, 3.8, 4.1, 4.3, 4.4, 4.5, 4.9, 10, 15	500
Riexinger et al., 2019 [25]	3T	100 ms	FB	24	0.2, 0.4, 0.7, 0.8, 1.1, 1.7, 3, 3.8, 4.1, 4.3, 4.4, 4.5, 4.9, 10, 15	500
Riexinger et al., 2021 ^c^ [26]	3T	45 ms	RT	16	0, 0.3, 1, 1.2, 1.5, 3.5, 5, 6	800

^a^ pilot study; ^b^ validation study; ^c^ in vivo study with optimized 16 b-values data set; No, number; NET, navigator-echo-triggering; FB, free breathing; RT, respiratory triggering; RT *, respiratory triggering using an air-filled pressure sensor fixed on the upper abdomen; b-value in s/mm^2^.

## Data Availability

The data presented in this study are available on request from the corresponding author. The data are not publicly available due to identity reasons.

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
