# Peer review of "Evidence of Tri-Exponential Decay for Liver Intravoxel Incoherent Motion MRI: A Review of Published Results and Limitations"

_diagnostics, 2021, doi:10.3390/diagnostics11020379_

Round 1
Reviewer 1 Report
It is an extensive review regarding imagistic methods to diagnose liver tumors and to differentiate solid tumors from metastases.
The paper is well documented, with proper and recent references.
The conclusions are supported by the results.
If possible, please also cite:
Gheorghe, Gina, A. P. Stoian, Mihnea-Alexandru Găman, B. Socea, Tiberiu Paul Neagu, A. Stanescu, O. Bratu, D. Mischianu, A. Suceveanu and C. Diaconu. “The Benefits and Risks of Antioxidant Treatment in Liver Diseases.” Revista De Chimie 70 (2019): 651-655.
Author Response
Thank you very much for your kind comments. As suggested, we added a citation of this article in the introduction.
"Gheorghe, Gina, A. P. Stoian, Mihnea-Alexandru Găman, B. Socea, Tiberiu Paul Neagu, A. Stanescu, O. Bratu, D. Mischianu, A. Suceveanu and C. Diaconu. “The Benefits and Risks of Antioxidant Treatment in Liver Diseases.” Revista De Chimie 70 (2019): 651-655"
Furthermore, the paper has been double checked by a native speaker for english language.
Reviewer 2 Report
Dear Authors,
the paper is a nice review about IVIM tri-exponential decay applied to liver.
It is well structured, with extensive and clear theoretical introduction. There is also a comprehensive discussion about b values selection, B0 field strength and echo time and their influence on tri-exponetial decay parameters.
The Authors also discuss about the limitations of this approach, like acquisition time and the difficulties in estimation of very fast component of diffusion.
The utility of this kind of model remains unclear but, as noted in the conclusions, further studies are necessary to investigate more this aspect of liver diffusion imaging.
I have nothing to add honestly.
Best Regards
Author Response
Thank you very much for your kind comments. This is a subject of great interest for us.
Furthermore, the paper has been double checked by a native speaker for english language.
Reviewer 3 Report
The review is focused on the scientific evidence of the existence of a more complex tri-exponential behavior, rather than the bi-exponential IVIM model, of the Diffusion MR signal allowing us to better describe the perfusion components in-vivo in different scenarios.
The paper is well written and the topic is for sure of interest to the readers.
In order to provide a more general message and to reach a wider public, I would suggest the authors to provide more general messages not focusing the discussion just on applications in the liver.
For example, as they discuss in paragrah 4.2, tri-exponential signal decay can be observed in other biological scenarios. I believe that this part is worthy of improvement providing more details as done with the liver.
For example, other recent papers were published for the kidney and could be used for this purpose:
- van der Bel, R., Gurney-Champion, O. J., Froeling, M., Stroes, E. S., Nederveen, A. J., & Krediet, C. P. (2017). A tri-exponential model for intravoxel incoherent motion analysis of the human kidney: in silico and during pharmacological renal perfusion modulation. European journal of radiology, 91, 168-174.
- Van Baalen, S., Froeling, M., Asselman, M., Klazen, C., Jeltes, C., Van Dijk, L., ... & Ten Haken, B. (2018). Mono, bi-and tri-exponential diffusion MRI modelling for renal solid masses and comparison with histopathological findings. Cancer Imaging, 18(1), 1-11.
Paragraph 4.3, regarding the model-fitting methods, should take into considerations other recent papers focused on the tri-exponential estimation of IVIM parameters. For example:
- Liu, J., Gambarota, G., Shu, H., Jiang, L., Leporq, B., Beuf, O., & Karfoul, A. (2017, December). Efficient sparsity-based algorithm for parameter estimation of the tri-exponential intra voxel incoherent motion (IVIM) model: Application to diffusion-weighted MR imaging in the liver. In 2017 IEEE 7th International Workshop on Computational Advances in Multi-Sensor Adaptive Processing (CAMSAP) (pp. 1-5). IEEE.
- Liu, J., Gambarota, G., Shu, H., Jiang, L., Leporq, B., Beuf, O., & Karfoul, A. (2018, October). All-in-one approach for constrained all-voxel tri-exponential IVIM model identification: Application to Diffusion-Weighted MR Imaging in the Liver. In 2018 52nd Asilomar Conference on Signals, Systems, and Computers (pp. 1192-1196). IEEE.
The estimation od Dfast/Dvfast parameters is usually poorly reliable in the IVIM approach. Bayesian methods have shown promising results in order to mitigate this limitation. The authors should cite some other recent papers that dealt with this problem:
- Lanzarone, E., Mastropietro, A., Scalco, E., Vidiri, A., & Rizzo, G. (2020). A novel bayesian approach with conditional autoregressive specification for intravoxel incoherent motion diffusion‐weighted MRI. NMR in Biomedicine, 33(3), e4201.
- Reliable estimation of incoherent motion parametric maps from diffusion‐weighted mri using fusion bootstrap moves. Med Image Anal. 2013; 17(3): 325‐ 336. , , , et al.
- Spatially‐constrained probability distribution model of incoherent motion (spim) for abdominal diffusion‐weighted MRI. Med Image Anal. 2016; 32: 173‐ 183. , , , , , .
Furthermore, even if the Bayesian approaches were just used, till now, for the bi-exponential model, the authors should try to speculate about their use in the tri-exponential case as this can probably represent an improvement of the state of the art. The same can be made considering the methods based on the ANNs.
Author Response
The review is focused on the scientific evidence of the existence of a more complex tri-exponential behavior, rather than the bi-exponential IVIM model, of the Diffusion MR signal allowing us to better describe the perfusion components in-vivo in different scenarios.
The paper is well written and the topic is for sure of interest to the readers.
Reply 1: Thank you very much for your comments.
In order to provide a more general message and to reach a wider public, I would suggest the authors to provide more general messages not focusing the discussion just on applications in the liver.
For example, as they discuss in paragrah 4.2, tri-exponential signal decay can be observed in other biological scenarios. I believe that this part is worthy of improvement providing more details as done with the liver.
For example, other recent papers were published for the kidney and could be used for this purpose:
- van der Bel, R., Gurney-Champion, O. J., Froeling, M., Stroes, E. S., Nederveen, A. J., & Krediet, C. P. (2017). A tri-exponential model for intravoxel incoherent motion analysis of the human kidney: in silico and during pharmacological renal perfusion modulation. European journal of radiology, 91, 168-174.
- Van Baalen, S., Froeling, M., Asselman, M., Klazen, C., Jeltes, C., Van Dijk, L., ... & Ten Haken, B. (2018). Mono, bi-and tri-exponential diffusion MRI modelling for renal solid masses and comparison with histopathological findings. Cancer Imaging, 18(1), 1-11.
Reply 2: Thank you very much for your comments. We discussed and cited these 2 articles in paragraph 4.2 (l. 226 – 239) and paragraph 4.9 (l. 497 – 508), respectively.
We preferred to focus on liver for different reasons. First, our staff works mainly on liver diseases. Second, since abdominal organs present different histological structures and physiological characteristics, the diffusion compartments that are assessed might be different. For instance, the liver presents a larger interstitial space (Disse space) than other abdominal organs and, as mentioned in Van der Bel’s article, pre-urine flow is present in the kidney. In addition, the vascular structure is quite different, with the liver presenting dual blood supply. Due to different contributions to the DWI signal of these diffusion compartments, the diffusion components that are detected by the tri-exponential model may not be the same. Also, due to different spins velocities, the optimal DWI protocol, the b-values distribution, might be different.
Paragraph 4.3, regarding the model-fitting methods, should take into considerations other recent papers focused on the tri-exponential estimation of IVIM parameters. For example:
- Liu, J., Gambarota, G., Shu, H., Jiang, L., Leporq, B., Beuf, O., & Karfoul, A. (2017, December). Efficient sparsity-based algorithm for parameter estimation of the tri-exponential intra voxel incoherent motion (IVIM) model: Application to diffusion-weighted MR imaging in the liver. In 2017 IEEE 7th International Workshop on Computational Advances in Multi-Sensor Adaptive Processing (CAMSAP) (pp. 1-5). IEEE.
- Liu, J., Gambarota, G., Shu, H., Jiang, L., Leporq, B., Beuf, O., & Karfoul, A. (2018, October). All-in-one approach for constrained all-voxel tri-exponential IVIM model identification: Application to Diffusion-Weighted MR Imaging in the Liver. In 2018 52nd Asilomar Conference on Signals, Systems, and Computers (pp. 1192-1196). IEEE.
Reply 3: Thank you very much for your comments. We discussed and cited these papers in paragraph 4.3 (l. 291 – 298).
The estimation of Dfast/Dvfast parameters is usually poorly reliable in the IVIM approach. Bayesian methods have shown promising results in order to mitigate this limitation. The authors should cite some other recent papers that dealt with this problem:
- Lanzarone, E., Mastropietro, A., Scalco, E., Vidiri, A., & Rizzo, G. (2020). A novel bayesian approach with conditional autoregressive specification for intravoxel incoherent motion diffusion‐weighted MRI. NMR in Biomedicine, 33(3), e4201.
- Freiman M, Perez‐Rossello JM, Callahan MJ, et al. Reliable estimation of incoherent motion parametric maps from diffusion‐weighted mri using fusion bootstrap moves. Med Image Anal. 2013; 17(3): 325‐ 336.
- Kurugol S, Freiman M, Afacan O, Perez‐Rossello JM, Callahan MJ, Warfield SK. Spatially‐constrained probability distribution model of incoherent motion (spim) for abdominal diffusion‐weighted MRI. Med Image Anal. 2016; 32: 173‐ 183.
Furthermore, even if the Bayesian approaches were just used, till now, for the bi-exponential model, the authors should try to speculate about their use in the tri-exponential case as this can probably represent an improvement of the state of the art. The same can be made considering the methods based on the ANNs.
Reply 4: Thank you very much for your comments. We discussed and cited these papers in paragraph 4.3 (l. 277 – 290). We also discussed the improvement that might be with Bayesian and ANNs methods.
Round 2
Reviewer 3 Report
The paper was improved with respect to the previous version. I still believe that a more general message, considering different anatomical areas, would have been an opportunity for the document but this review deserves publication even in this form.
I have nothing else to add.